# Pediatric Cervicofacial Necrotizing Fasciitis—A Challenge for a Medical Team

**DOI:** 10.3390/children10071262

**Published:** 2023-07-22

**Authors:** Adina Simona Coșarcă, Dániel Száva, Bálint Bögözi, Alina Iacob, Anca Frățilă, Guzun Sergiu

**Affiliations:** 1Department of Oral and Maxillo Facial Surgery, George Emil Palade University of Medicine, Pharmacy, Science and Technology of Targu Mures, 540142 Targu Mures, Romania; adina.cosarca@umfst.ro (A.S.C.); balint.bogozi@umfst.ro (B.B.); alina.iacob@umfst.ro (A.I.); 2Oral and Maxillo Facial Surgery Clinic, Emergency County Hospital Târgu Mures, Gheorghe Marinesscu Street, No. 50, 540136 Targu Mures, Romania; ancamagaliuc@gmail.com (A.F.); guzunmatei@gmail.com (G.S.)

**Keywords:** pericoronitis, necrotizing fasciitis, multidisciplinary approach, odontogenic infection

## Abstract

Cervical necrotizing fasciitis is a very rare complication of a bacterial infection that can have a dental cause. This type of infection typically affects fascial plane, which has a poor blood supply and can affect soft tissue and cervical fascia and can spread quickly causing infection of mediastinum. Initially, in the first stage, the overlying tissues are unaffected, and this can delay diagnosis and surgical intervention. Incidence in children is extremely rare and can be frequently associated with various other general pathologies that decrease the immune system response. We present a case of a young 12-year-old boy diagnosed with this type of infection in the head and neck as a complication of a second inferior molar pericoronitis. The treatment and the management of the case was difficult not only due to the presence of the infection but also because of the prolonged intubation.

## 1. Introduction

Necrotizing fasciitis (NF) is a severe infection of the soft tissues, fasciae and subcutaneous tissues, even the muscles. It is characterized by a rapid and fulminant progression of rapidly spreading necrosis, which can even put the patient in a life-threatening condition [1]. Infections in NF are not limited and are characterized by the presence of inflammation and necrosis, extending in depth. In the affected areas, the skin may have a normal aspect. It is a rare infectious entity, with a difficult diagnostic and therapeutic management for a surgeon [2,3,4]. It has been reported in 0.03% cases as the cause of hospitalization [5], i.e., 0.08 per 100,000 children/year [4]. NF is more common in middle-aged adults (about 40 years old), without gender, race or geographic predilection [6]. Most injuries are reported at the trunk level (umbilical region, inguinal-perineal region after circumcision and limbs), and the cervicofacial region being much less often involved [7]. The most commonly affected cervicofacial spaces are submandibular, sublingual, submental, lateropharyngeal and retropharyngeal with cervical extension [8]. Microbiologically, NF has been classified as either type 1 (polymicrobial) or type 2 (monomicrobial). Polymicrobial infections are more common, with cultures yielding a mixture of aerobic and anaerobic organisms. These infections typically occur in the perineum and trunk. Type 1 NF occurs in immunocompromised individuals, such as patients with diabetes mellitus or chronic renal failure. Monomicrobial infections are less common than the polymicrobial variety. These typically occur in the limbs and affect healthy patients with no implicative comorbidities. S. pyogenes and S. aureus are the usual pathogens. Type 2 NF might be associated with toxic shock syndrome [9]. In head and neck region, the most commonly reported predisposing factor for NF is a primary odontogenic infection or a post-tooth extraction infection [10,11,12]. Complicated caries and dental gangrene are the causes of two-thirds of cases, followed by pericoronitis and periodontal disease [13]. Tonsil infections, salivary gland infections, otogenic and dermatological infections are other causes. Predisposing factors vary according to age, diabetes, obesity, malnutrition, immunocompromised state, peripheral vascular diseases and drug use [14,15,16]. Nonsteroidal anti-inflammatory drugs have also been implicated as a predisposing factor [17]. In children, NFs are frequently misdiagnosed at first presentation to the doctor, most of the time they are diagnosed as simple soft tissue infections such as cellulitis, upper respiratory tract infections and cervical lymphadenitis, and this leads to delayed treatment [4].

For the management of necrotizing fasciitis, a multidisciplinary approach is mandatory; the multidisciplinary team consists of OMF, thoracic, pediatric and vascular surgeons and ENT specialists. Early surgical debridement and early initiation of antibiotic treatment are the most important therapeutic measures. The most commonly used antibiotic treatment can be a combination of aminoglycosides or third-generation cephalosporin with clindamycin [4]. Early and extensive surgical debridement is a widely accepted clinical approach and the mainstay of effective treatment. The goals of surgery are debridement and removal of all the necrotic tissue, including muscle, fascia and skin and to control the progression of NF.

We therefore summarize our experience with a 12-year-old boy diagnosed with an aggressive cervicofacial NF, and we focus our attention on the importance of an early diagnosis and treatment.

## 2. Case Presentation

A 12-year-old boy from an urban area presented to Dental Emergency Room of Emergency County Hospital Târgu Mures, complaining of discomfort and pain located at tooth 3.7. Congestive pericoronitis at tooth 3.7 was suspected. Local irrigations were performed using antiseptic solution—chlorhexidine gluconate 0.2%—under the operculum covering the tooth 3.7. Painkillers and anti-inflammatory drugs were prescribed. After 48 h, the symptomatology worsened, and the child presented again at the Dental Emergency Room, complaining of radiating pain in the left side of the mandible that could not be accurately located. The patient was diagnosed with suppurative pericoronitis at tooth 3.7. Local irrigation with chlorhexidine gluconate 0.2% was performed, and peroral antibiotic treatment was prescribed (Amoxicillin tablets).

After 12 h, the general state of health of the patient worsened, increased pain on the left side of the face and facial asymmetry due to edema of the left submandibular region extending towards the left submental region were present. Local hyperemia of the skin of the submandibular region was observed, and on palpation, the region had firm consistency, and it was painful but presented no fluctuation. Thorough examination of the oral cavity was difficult to perform, due to the presence of the trismus, but significant edema on the anterior and left side of the floor of the mouth could be seen. The patient was admitted as an emergency case to the Oral and Maxillofacial Surgery Department with a diagnosis of left submandibular abscess. A general physical examination was performed. Class 1 obesity, oscillating blood sugar levels and increased blood pressure were noticed, but without any other previously diagnosed general illness. The emergency treatment was performed under local anesthesia and consisted of incision, evacuation and drainage of the left submandibular abscess. Dissection of the left submandibular space was performed using Hilton technique. Drainage tubes were positioned and sutured in the affected anatomical space. Despite the emergency surgical treatment and the associated intravenous antibiotic medication treatment initiated (Ceftriaxone and Metronidazole), the evolution of the patient status was unfavorable. After 24 h, the first surgical intervention left submandibular edema expansion towards the left submental area, towards the midline and infiltration of the contralateral submandibular and submental spaces, swallowing difficulties and worsening of the patient’s general condition suggested a second surgical exploration of these spaces. Right submandibular and submental spaces were drained, and bacteriological examination was requested. During the second intervention, the presence of gas in the right submandibular region was noticed with the presence of necrotic and fetid odor secretion. The patient was transferred to the ICU (Intensive Care Unit). Two hours after the transfer, the general health status of the patient worsened. The patient showed signs of psychomotor agitation and became uncooperative and was sedated and intubated. Postoperative head, neck and thorax CT scan was performed in order to evaluate the extension of the lesions.

The contrast-enhanced CT scan of head, neck and thorax revealed multiple mixed fluid-air densities in the infratemporal, sublingual, submandibular, submental, parapharyngeal, retropharyngeal and peripharyngeal spaces and extended to the mediastinal level, on the upper and middle floors of the anterior and posterior compartment; it also revealed edematous infiltration of perimandibular, submandibular and laterocervical subcutaneous bilateral soft tissues (Figure 1).

On the same day, surgical reintervention was performed by joining and widening the bilateral subangulomandibular incisions, and the submandibular, sublingual, the base of the tongue and bilateral infratemporal fossae were opened and explored by means of blunt dissection. Necrotic secretions and fetid-smelling gases were eliminated. A right curved horizontal laterocervical incision was performed to open the vascular space of the common carotid artery and its branches, the internal jugular vein, the paratracheal space (anterior, posterior and right lateral), reaching also to the right prevertebral space. Drainage of the anterior superior mediastinum was performed, entering from the neck incision, dissecting bilaterally along the anterior border of the SCM muscle, incising the superficial cervical fascia and both layers of the deep cervical fascia above the jugular notch. Keeping contact with the internal surface of the sternal manubrium, blunt dissection was made, approximately 12 cm long, caudally. Necrotic secretions were evacuated, and drainage was achieved with silicone tube. (Figure 2). All the anatomic spaces described previously were meshed. Postoperative CT scan was made in order to evaluate the drainage of all anatomical spaces involved (Figure 2). After the surgery, the treatment continued in the children’s Intensive Care Unit, and the patient remained sedated, intubated and mechanically ventilated. The wound dressings were changed three time/day. During daily wound care surgical debridement, necrotic fascia mainly around the great vessels of the neck were excised (with skeletonization of the common carotid artery and its internal and external branches, internal jugular vein, right next to the right brachiocephalic artery). Due to the complex local treatment, local evolution was slowly favorable.

Regarding the general status of the patient, during the first days spent in the ICU, the patient presented acid–base and electrolyte imbalances, having a pH fluctuating between 7.26–7.56 (n = 7.3–7.45). Bacteriological examination of secretions collected from intraoperative wounds revealed the presence of Streptococcus anginosus and Granulicatella adiacens.

After 48 h from the third surgical intervention, control CT scan of head, neck and chest with contrast was performed. It revealed a quantitative reduction in air densities in the laterocervical, submandibular, parapharyngeal and mediastinal spaces, but with the presence of nonfluid collections in the cervical and mediastinal spaces having approximately the same aspect as in the previous CT scan; minimal bilateral postero-basal pleural fluid collection was observed.

Entering from the neck scar, blunt dissection was made from the right side of the neck in order to drain the prevertebral collection that descends into the posterior mediastinum and necrectomy of the necrotic tissues and fasciae was performed.

On 4th day in the ICU, signs of peripheral vascular damage such as severe acrocyanosis and pressure ulcers in the calcaneus and leg regions were observed.

On the 5th day in the ICU, during dressing of the wounds (at 9:00 P.M.), a massive erosive hemorrhage occurred from the right laterocervical postoperative wound. The bleeding was stopped by applying direct pressure to the wound.

Head, neck and chest CT angiography was performed, and no source of active bleeding in the thoracic cavity was revealed.

During the 7th day in ICU, the postoperative wounds were reviewed. Local evolution was favorable, and fascial necrosis and pathological secretions were absent. Granulation tissue is present in the wound. These aspects led us to the decision of finally closing the wound (wound secondary sutures) and of suppressing the remaining drain tube located in the anterior mediastinum (Figure 3).

As the general health status did not improve on the 19th day in the ICU, it was decided to reevaluate the case by performing a new CT scan of head, neck and chest. The CT scan revealed a fused retrosternal collection from the anterior cervical region having the dimensions of 11/45/73 mm (AP/LL/CC), thrombosis of the right internal jugular vein, bronchopneumonia (presence of condensation foci in all pulmonary lobes located subpleurally) and also the presence of posterior basal atelectasis. Clinical laboratory test results showed a higher number of white blood cells (leukocytosis) and PCR and lactate dehydrogenase (WBC—39. 103/μL; PCR—350 mg/L; LDH—125–220). These clinical data led us to the therapeutic decision to review the postoperative right laterocervical wound and the anterior mediastinum. A mixed team of maxillofacial and thoracic surgeons was formed. During the surgical exploration of the anterior mediastinum, purulent collections were not detected, only inflammatory infiltrate. The general health status of the patient worsened due to pulmonary hemorrhage complicated with bronchopneumonia.

On 25th day in the ICU, bronchial lavages were performed (for 5 days), with the removal of blood clot formed in larger caliber bronchi (Figure 4).

The general treatment of the patient was complex and was permanently adjusted to the curative needs, combining rheological treatment, Dopamine (positive ionotropic support), diuretics, hypotensive drugs, Clexane, hepatoprotective drugs, antibiotics, anti-inflammatory, analgesic, corticosteroid, vitamin therapy, fresh frozen plasma (FFP) and blood transfusion.

The initial antibiotic treatment during hospitalization was Ceftriaxone and Metronidazole. After the patient was transferred to the ICU, the initial antibiotic therapy was replaced by a combination of Meropenem, Vancomycin and Metronidazole. In the ICU because of uncertain local evolution, various combinations of antibiotic treatment therapies were applied: Meropenem, Vancomycin and Clindamycin; Later, Clindamycin, Gentamicin, Imipenem and Linezolid; and finally, a combination of Gentamicin, Imipenem and Linezolid. During the ICU hospitalization, a nasogastric tube was placed for parenteral nutrition.

On 27th day in the ICU, tracheostomy was performed due to need of prolonged mechanical ventilation and respiratory recovery.

On 34th day in the ICU, general condition of the patient stabilized and slowly improved, confirmed by the improvement of the clinical laboratory test results. Final CT scan of the head, neck and thoracic cavity excluded any other pathologies. (Figure 5). Extubation and tracheostomy tube was removed, and the patient had adequate spontaneous breathing.

On 39th day, the patient was transferred to the OMF Surgery Department for further specialized treatment. Due to prolonged immobilization, the patient also exhibited pressure ulcers (occipital, legs and calcaneus), showing signs of healing indicated by the presence of granulation tissue and scarring (Figure 6 and Figure 7).

A Pediatric Neuropsychiatry Consult was also performed after which the diagnosis of acute flaccid paralysis and anxiety disorder was established. When the general state of health was stable, an OPG was performed, and on the tooth 3.6, a deep cavity damaging the pulp chamber was observed, and a periapical reaction was evident at the distal root (Figure 8a). Intraorally, we observed that the operculum of tooth 3.7 was partly present on the distal occlusal surface (Figure 8b). It was decided to extract tooth 3.6 and curettage the periapical lesion (Figure 8c).

On 48th day, the patient was discharged with the following recommendations: kinetic therapy, specific treatment according to recommendations of Pediatric Neuropsychiatry, dressing of all pressure ulcers and post-tracheostomy wound and periodic check-ups. Two weeks after discharge from the hospital, spontaneous closure of the tracheostomy wound and postoperative scars on the neck were observed (Figure 9).

## 3. Discussion

Pathology of cervicofacial infection in children is frequently associated with dental pathology and is most often represented by lymphadenitis.

The clinical examination is difficult to perform in case of a child, due to strain that can be generated by the young patient’s meeting with a doctor. The anamnesis is most often inconclusive because the child has difficulty describing their painful sensations and cannot accurately specify their location. This makes it difficult for the doctor to establish the correct diagnosis. In the age group of 6–12 years, the dental eruption of permanent molars occurs. Development of dental caries during this period and their rapid progression to the pulp chamber, which can be complicated with pulp gangrene, can affect the tooth having incomplete root formation, thereby stopping root development. Consequently, the root canal is large, with thin and fragile walls and an open apex [18]. And this is a quick exit gate for bacteria into the surrounding spaces. Eruption of the molars can also be associated with another type of pathology, namely, pericoronitis (inflammation of the mucous membrane above the molars). The two associated pathologies can lead to multiplication of bacteria, exacerbation of pain and implicitly a much faster diffusion into neighboring anatomical spaces (submandibular, sublingual, submental, prevertebral, parapharyngeal and retropharyngeal spaces).

In our case, we consider the pericoronitis of the lower left second molar as the starting point of the infection. The accumulation of bacteria under the mucous cap could go unnoticed due to the vague symptomatology at the time of presentation to the dentist. At the time of presentation to the dentist, the clinical examination does not highlight other teeth having dental caries or coronary destruction. The X-ray showed a dental filling present in teeth 3.6. The coronal filling of tooth 3.6 was performed using physiognomic composite materials that could eventually cause pulp necrosis followed by pulp gangrene at tooth 3.6. This could also be another cause of infection. Since the distal root of the lower second molar is located under the insertion of the mylohyoid muscle, the spread of the infectious process occurred quickly towards the left submandibular space, and from this point, the infection can extend quickly to the opposite side. The bacteriological examination showed the presence of Streptococcus anginosus and Granulicatella adiacens. The presence of Granulicatella adiacens in the cervical level can explain the rapid evolution of the case from submandibular abscess to necrotizing fasciitis within a few hours of admission. Granulicatella adiacens has been described by others studies as being involved in endocarditis infections, breast or peritoneal implants [19].

Streptococcus anginosus colonizes the mucous membranes of the oral cavity and in some cases blood stream infections. Particularity of this case is that despite the administration of combined wide spectrum intravenous antibiotic treatment from the day of the admission (Ceftriaxone and Metronidazole), propagation and dissemination of the septic process was extremely fast. In approximately 48 h, it diffused from the left submandibular space towards the contralateral side, infratemporal fossa and the mediastinum. At this moment, we did not dispose of the result of the bacteriological examination. Theoretically, Streptococcus anginosus is susceptible to the combination of Ceftriaxone and Metronidazole. Granulicatella adiacens is only partly susceptible to this antibiotic combination, and the synergic activity of these two aggressive strains of bacteria might have contributed to the fulminant evolution of the septic process. Dental infections are known to be polymicrobial, and there were no bacteriological examinations for anaerobic strains conducted, so this aspect of the investigations had limitations. In cases of cervical and facial infections of the deep regions, the CT scan is the only reliable paraclinical examination that can show the presence of pus collection and necrosis (gases) in these areas. Clinical signs are always blurred by the presence of the soft cervical and facial tissues, in particular in case of obese children having “short neck” and abundant adipose soft tissue. When compared with adults, in case of children, the anatomic landmarks between head and neck region are less concrete due to smaller chin, more caudal rotation of the mandible and the shorter submental and submandibular regions.

For cervicofacial necrotizing fasciitis, the treatment of choice is surgical, and it is a major emergency because it is a rare, fatal, rapidly progressive soft tissue infection. After extensive surgery, antimicrobial treatment was changed to Meropenem, Vancomycin and Metronidazole. This combination should have covered all the bacterial strains involved in the infection. It would be optimal to have a targeted antibiotic therapy, but to obtain bacteriological examination results is time consuming, and in these cases, empirical antibiotic therapy is initiated, which is later adjusted to the bacteriological examination. In this case, it consisted of antibiotic combinations such as Clindamycin, Gentamicin, Imipenem and Linezolid and finally a combination of Gentamicin, Imipenem and Linezolid

The surgical treatment of necrotizing fasciitis at the cervical region is difficult to achieve due to the presence of important vascular anatomical elements at this level and their sclerosis by the necrotic process. Repeated necrectomies, irrigation with antiseptic solutions and applications of antiseptics can mobilize the clots formed and can lead to massive, even fatal hemorrhages. We faced such a situation in our clinical case; during local debridement of the necrotic tissue, significant bleeding started from the superior thyroid artery; the bleeding was stopped by local hemostasis. A laterocervical infection can rapidly progress to the mediastinum, causing mediastinitis. This is a serious complication having a high mortality rate in the absence of treatment or late treatment, especially among patients with other associated pathologies such as diabetes, heart failure and obesity. In our presented clinical case, the 12-year-old patient exhibited class 1 obesity, fluctuating blood sugar levels and increased blood pressure values during hospitalization. Due to multiple comorbidities, this clinical case could have an unfavorable even fatal prognosis.

Other particularities of this case were pulmonary complications due to prolonged intubation and mechanical ventilation. It is known that the bronchopneumonia development rate is highest in the first 2 weeks of intubation [20]. Among the risk factors that may lead to bronchopneumonia early on from the first 48 h of intubation, are uncontrolled infection, oral cavity infection and sepsis, swallowing difficulties and aspiration complications, which were present in this case [21,22]. Ventilation-assisted bronchopneumonia as a mortality attributable has been reported in up to 40% of the cases [23]. Diagnosis of ventilation-assisted bronchopneumonia in these patients is difficult. The positive diagnosis requires multiple criteria to be fulfilled, such as new and persistent (>48 h) infiltrate in chest radiograph plus two or more of the three criteria: (1) fever of >38.3 °C, (2) leukocytosis of >12 × 109/mL, and/or (3) purulent tracheobronchial secretions with a sensitivity of 69% and a specificity of 75%, for establishing the diagnosis of ventilation-assisted pneumonia [24]. Particularly in this case, leukocytosis and fever were present due to the extensive neck infection, without the presence of tracheobronchial secretions. Initial attempt of extubation was made after the control of the neck infection failed, and the patient presented with respiratory insufficiency and had to be reintubated. After long-term endotracheal intubation postextubation, supraglottic and infraglottic aspiration and subsequent aspiration, pneumonia occured. In case of pediatric patients, postextubation dysphagia was present in 29–69% of the studied cases, with an increase of 1.7% for every hour of intubation [25,26,27]. Beside this complication, particularly in this case, dysphagia could be also caused due to the extensive necrosis and consecutive debridement, surgical intervention of the neck and perioral spaces. Because of these multiple overlapping risk factors such as ventilation-assisted bronchopneumonia and postextubation dysphagia causing aspiration pneumonia, the general status of the patient worsened. Bronchoscopy identified multiple blood clots in the major bronchioles and were removed with repeated bronchial lavages. This opened another question, namely, the pulmonary hemorrhage problem. In our understanding, in this case, the cause of pulmonary hemorrhage could have been the severe negative pulmonary pressure. Excessive negative pulmonary pressure builds up in the lungs due to upper airway obstruction, which can lead to mechanical stress on the pulmonary capillaries, causing hemorrhage. Although negative pulmonary edema/hemorrhage seems to appear rarely in case of children, it is a major, life-threatening situation [28]. Otherwise, negative pressure pulmonary edema and hemorrhage is specific in young and healthy population and can exert high pressure on the chest wall [29]. Several causes of negative pressure pulmonary edema have been identified, most common being laryngospasm after extubation in 50% of the identified cases [30] then mandible fracture open reduction [31], intubation tube obstruction, nasal fracture reduction [32] and other interventions. It appears rarely in case of children and even more rarely appears with pulmonary hemorrhage.

Time spent in the ICU, prolonged bed rest and even using the specific methods of mobilization for the intubated patient can lead to appearance of pressure ulcers on the extremities, which influence and affect the complete physical and mental healing of a pediatric patient.

Although specific mobilization was carried out every 4–5 h and anti-bedsore mattress was used, due to the peripheral vascular damage, aggravated by metabolic ketoacidosis and septic shock, pressure ulcers evolved.

The management of such a case involves an interdisciplinary team and does not end when the patient is discharged from the hospital. Since the local evolution of the patient can change drastically every hour, the patient’s supervision is indicated to be performed by the same initial medical team for right monitoring and evaluation of the clinical case.

Necrotizing fasciitis is rare in the head and neck region, but it is potentially life threatening, most frequently caused by the second and third lower molars in case of adults [33], and it can appear in case of children and can be caused by the periapical pathology of the first molar.

## 4. Conclusions

Oral cavity infections can have a fulminant evolution, having a rather serious prognosis and can be a life-threatening development. In case of children, most common odontogenic cause is dental decay complicated with pulp gangrene and periapical abscess. Rapid development, particularly in case of the infection of the deep perioral spaces such as the submandibular, sublingual, parapharyngeal and pterigomaxillar spaces, is usually possible for the posterior permanent teeth, most frequently from the first and second molars, which erupt from the age of 6 years to 12 years. In case of children, young molars have long roots, inserted deep in the jaws, and large root canals, with wide apical foramen, and permit rapid development of the deep periapical abscess. Usually dental treatment of these molars is laborious, due to the poor collaboration with children. Silent oligosymptomatic dental pulp necrosis and gangrene under large dental fillings is also common in case of children due to large dental pulp chamber in case of recently erupted molars, which lay in the proximity of the dental filling material. Necrosis and gangrene can also occur due to cytotoxic properties of the dental filling materials. Without dental X-ray and thorough dental examination, no correct dental emergency treatment can be implemented. In most cases, a dental trepanation at the visit to the dentist can prevent the later complications. Antibiotic treatment without proper dental treatment contributes to the evolution of multi-resistant anaerobic strains.

Involvement of the deep perioral spaces such as the submandibular space causes severe trismus, which obstructs any other dental treatment for the period of its persistence. Intraoral examination is also limited due to intense trismus, and poor collaboration with children also makes the treatment more difficult.

In order to avoid difficult intubation with potential risks, submandibular abscess incision with Hilton technique is performed under sedation, which permits only limited surgical access and exploration time, and can lead to insufficient drainage of the affected space and limited treatment possibilities. Bacteriological examination results are time consuming, and in case of rapid, unfavorable, extensive infections, with no tendency of abscess collection and limited to a single anatomic space, wide spectrum intravenous empirical antibiotic treatment must be initiated.

Usually children, due to lack of understanding, cannot describe exactly the subjective symptomatology. Deep cervical and facial infection remain clinically masked for longer periods of time, in particular in case of children, where anatomy of the head and neck and specific landmarks are different than in adults. Obesity and abundant adipose soft tissues blur the clinical image even more. CT scan of the head, neck and even the thorax in case of severe infections is the most reliable imaging procedure that can indicate the emergency surgical treatment. Surgical treatment has to be extensive, and control CT scan should be performed in order to confirm the exploration and the drainage of the involved anatomic spaces.

Until the result of the microbiological exam are available, empirically used combination of wide spectrum antibiotic therapy should be implemented, and after results are available, targeted antibiotic therapy should be implemented.

Leaving the surgical wound open for longer periods of time and oxygenation of the involved deep anatomic regions can assure effective treatment against anaerobic strains. Repeated necrectomies consisting of the excision of fascia can lead to lesions on the blood vessels of the head and neck, leading to uncontrollable bleeding.

Periodic reevaluation of the local and regional evolution, brain edema and brain abscess should be excluded using repeated CT scans.

Long intubation period can cause ventilation-assisted bronchopneumonia, aspiration pneumonia and negative pressure edema and hemorrhage, specifically in case of young adults.

Ketoacidosis and septic shock contribute to pressure ulcers in case of prolonged bedrest.

Multidisciplinary treatment has to be implemented as immediate treatment and after-treatment.

## Figures and Tables

**Figure 1 children-10-01262-f001:**
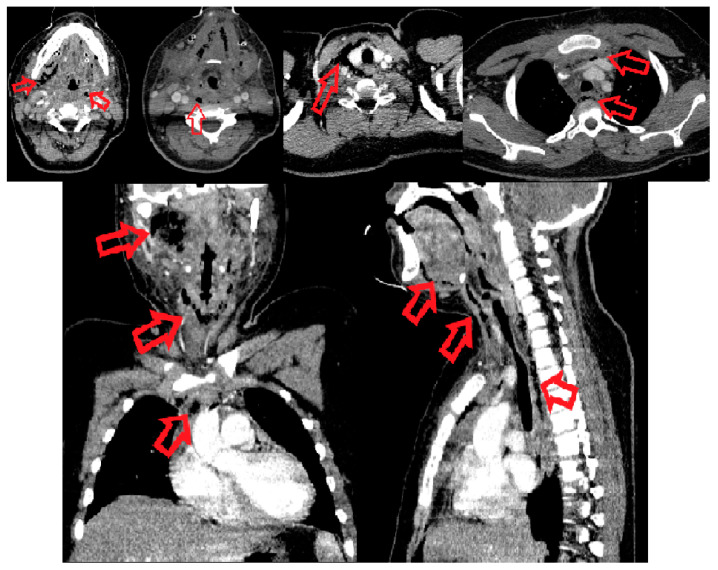
CT scan. The red arrows point to the necrotic and gaseous collections in the affected areas—infratemporal, sublingual, submandibular, submental, parapharyngeal, retropharyngeal and peripharyngeal spaces, and extend to the anterior upper mediastinum.

**Figure 2 children-10-01262-f002:**
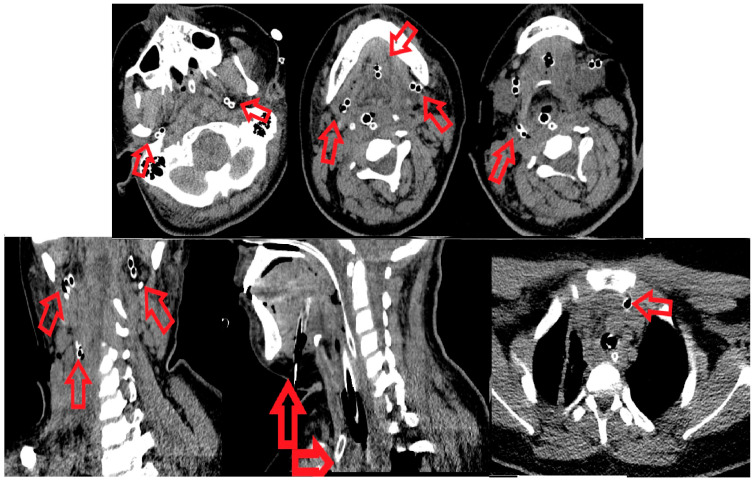
CT scan: Postoperative reevaluation of the involved anatomical spaces. Red arrows point to the drainage tubes inserted into the infratemporal region bilaterally, submandibular space bilateral, floor of the mouth, parapharyngeal and retropharyngeal space and anterior mediastinum.

**Figure 3 children-10-01262-f003:**
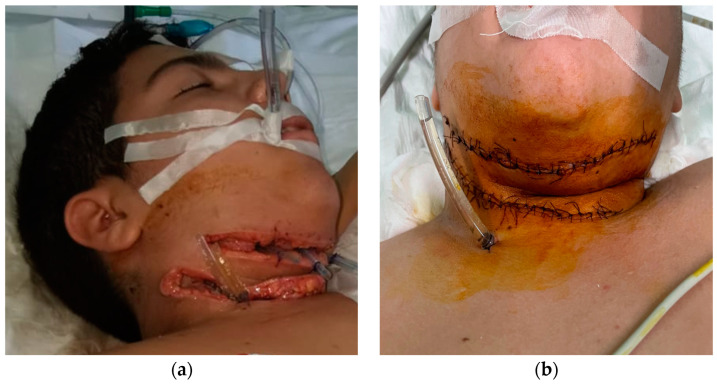
(**a**) Absence of the local fascial necrosis and pathological secretions and (**b**) wound secondary sutures and the last drain tube located in the anterior mediastinum that was removed.

**Figure 4 children-10-01262-f004:**
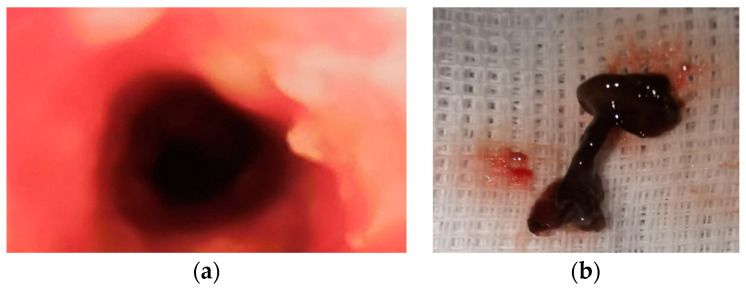
Blood clot: (**a**) endoscopic view and (**b**) removal of the blood clot.

**Figure 5 children-10-01262-f005:**
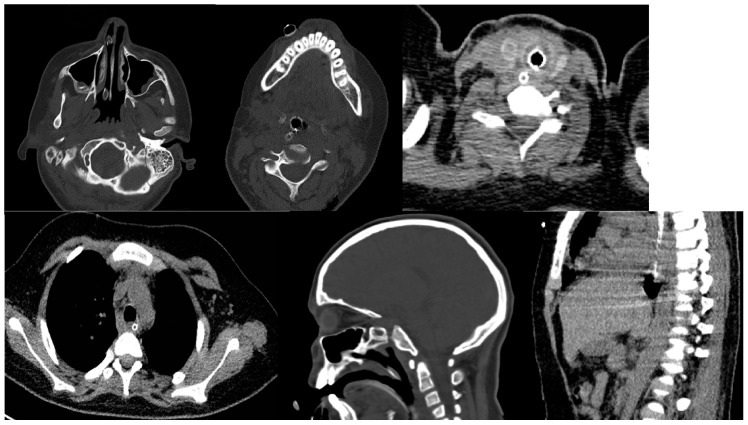
Final CT scan showing healing of the head, neck and mediastinum, with no further pathologies.

**Figure 6 children-10-01262-f006:**
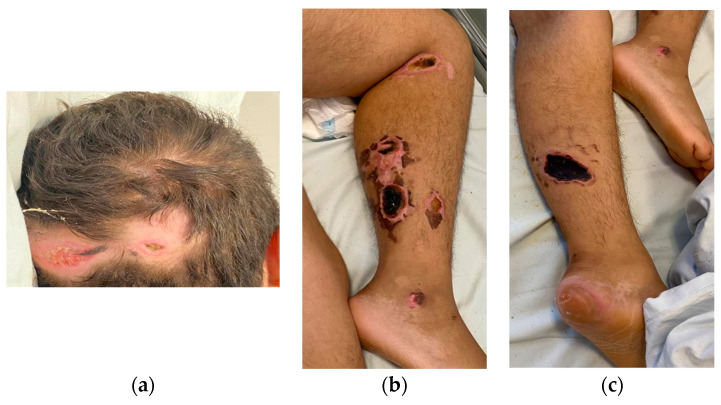
Pressure ulcers: (**a**) occipital, (**b**) legs and (**c**) calcaneus (heel).

**Figure 7 children-10-01262-f007:**
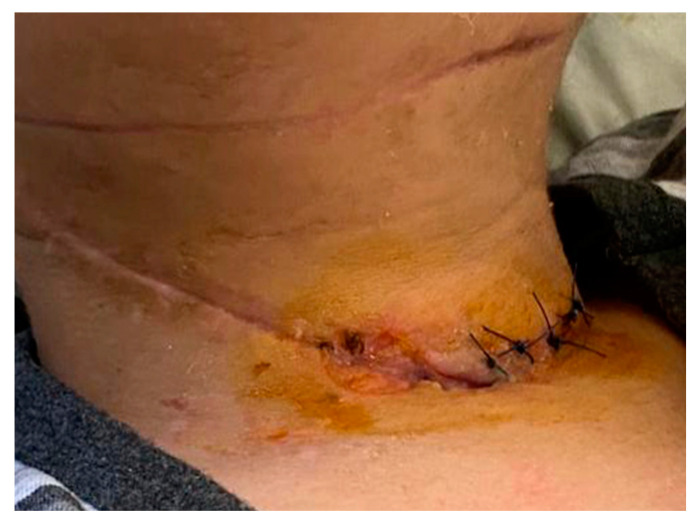
Postoperative wound during healing.

**Figure 8 children-10-01262-f008:**
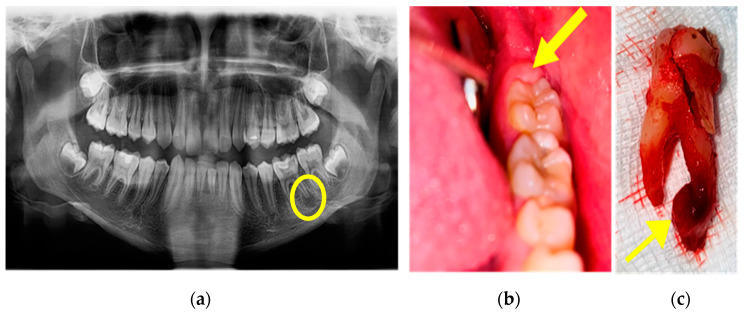
(**a**) OPG showing periapical lesion on distal root of tooth 3.6, (**b**) the mucous cap on 3.7 (coronal distal part), and (**c**) extraction of tooth 3.6 and the periapical lesion.

**Figure 9 children-10-01262-f009:**
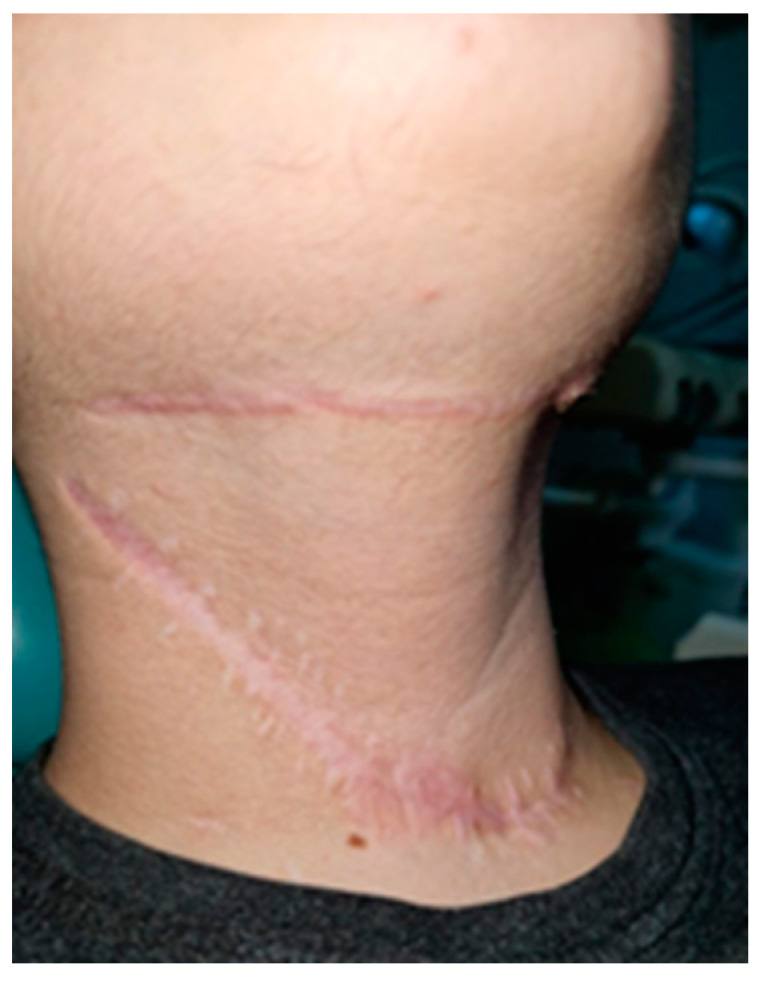
Postoperative scars of the neck.

## Data Availability

Not applicable.

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
