# Peer review of "Pediatric Cervicofacial Necrotizing Fasciitis—A Challenge for a Medical Team"

_children, 2023, doi:10.3390/children10071262_

Round 1
Reviewer 1 Report
In this manuscript, the authors present a case report on Pediatric Cervicofacial Necrotizing Fasciitis. There are several areas that can be improved to address certain concerns:
1. To enhance clarity and facilitate understanding of the patient's treatment progress, I suggest that the authors combine multiple CT scan results into a single figure. Additionally, indicating the specific regions that changed during the treatment process would provide a visual representation of the patient's condition over time.
2. In line 279-280, the presence of Streptococcus anginosus and Granulicatella adiacens is mentioned. It would be helpful to provide information on whether any specific treatments were applied to target these two pathogens. Elaborating on the treatment approach for these specific pathogens would enhance the understanding of the therapeutic strategy employed.
3. The authors should expand the conclusions section to provide a clearer understanding of the lessons learned from this case. It would be valuable to highlight any efficient methods or interventions utilized during the treatment of this case that could be applicable to future cases. This expansion would provide practical insights for clinicians and researchers in similar scenarios.
4. I recommend that the authors thoroughly review the captions for all figures in the manuscript. I noticed instances of incomplete content in some captions, such as Figure 8. Furthermore, there was an error in figure naming, as mentioned in line 229, where it should be Figure 6 instead of Figure 1. Reviewing and revising the figure captions will ensure accurate and comprehensive descriptions of the figures, improving the overall clarity of the manuscript.
By addressing these concerns, the authors can enhance the manuscript and provide a more comprehensive understanding of the case, its treatment, and the potential implications for future clinical practice.
Extensive editing of English language required
Author Response
Response to Reviewer 1 Comments
Point 1: To enhance clarity and facilitate understanding of the patient's treatment progress, I suggest that the authors combine multiple CT scan results into a single figure. Additionally, indicating the specific regions that changed during the treatment process would provide a visual representation of the patient's condition over time.
Response 1: The contrast-enhanced CT scan of head, neck and thorax revealed multiple mixed fluid-air densities in the infratemporal, sublingual, submandibular, submental, parapharyngeal, retropharyngeal, peripharyngeal spaces and having extension to the mediastinal level, on the upper and middle floors of the anterior and posterior compartment; also revealed edematous infiltration of peri-, submandibular and laterocervical subcutaneous soft tissues bilateral (Figure 1).
Figure 1: CT scan . The red arrows pointing to the necrotic and gaseous collections in the affected areas- infratemporal, sublingual, submandibular, submental, parapharyngeal, retropharyngeal, peripharyngeal spaces and having extension to the anterior upper mediastinum.
In the same day surgical reintervention was performed by joining and widening the bilateral subangulomandibular incisions, the submandibular, sublingual, the base of the tongue and bilateral infratemporal fossae were opened, explored by means of blunt dissection. Necrotic secretions and fetid-smelling gases were eliminated. A right curved horizontal laterocervical incision was performed to open the vascular space of the common carotid artery and its branches, the internal jugular vein, the paratracheal space (anterior, posterior and right lateral), reaching also to the right prevertebral space. Drainage of the anterior superior mediastinum was performed, entering from the neck incision, dissecting bilaterally along the anterior border of the SCM muscle, incising the superficial cervical fascia, and both layers of the deep cervical fascia above the jugular notch. Keeping contact with the internal surface of the manubrium sterni, blunt dissection was made, approximately 12 cm long, caudally. Necrotic secretions were evacuated, and drainage was achieved with silicone tube. (Figure 2). All the anatomic spaces described previously were meshed. Postoperative CT scan was made in order to evaluate the drainage of all anatomical spaces involved (Figure 2). After the surgery, the treatment continued in the children's intensive care unit, the patient remained sedated, intubated and mechanically ventilated. The wound dressings were changed 3 time/day. During daily wound care surgical debridement, necrotic fascia mainly around the great vessels of the neck were excised(with skeletonization of the common carotid artery and its internal and external branches, internal jugular vein, right next to the right brachiocephalic artery). Due to the complex local treatment, local evolution was slowly favorable.
Figure 2: CT scan: Postoperative reevaluation of the involved anatomical spaces. Red arrows pointing to the drainage tubes inserted to the infratemporal region bilaterally, submandibular space bilateral, foor of the mouth, para and retropharyngeal space and anterior mediastinum
Regarding the general status of the patient, during the first days spent in the ICU, the patient presented acid-base and hydroelectrolyte imbalances, having oscillating pH between 7.26 – 7.56 (n = 7.3 – 7.45). Bacteriological examination of secretions taken from intraoperative wounds revealed the presence of Streptococcus anginosus and Granulicatella adiacens.
48 hours after the third surgical intervention, control CT scan of head, neck and chest with contrast was made. It revealed a quantitative reduction of air densities in the laterocervical, submandibular, parapharyngeal and mediastinal spaces, but with the presence of parafluid collections in the cervical and mediastinal spaces having approximately the same aspect as in the previous CT scan; minimal bilateral postero-basal pleural fluid collection was observed.
Entering from the neck scar, blunt dissection was made from the right side of the neck in order to drain the prevertebral collection that descends into the posterior mediastinum and necrectomy of the necrotic tissues and fascia was performed.
On 4th day in ICU: signs of peripheral vascular damage - severe acrocyanosis and pressure ulcers in the calcaneus and legs regions were observed.
On the 5th day in ICU, during dressing of the wounds (hour 9:00 P.M.) a massive erosive hemorrhage from the right laterocervical postoperative wound was produced. The bleeding was stopped by applying direct pressure to the wound.
Head, neck, chest CT angiography is performed and no source of active bleeding in the thoracic cavity was revealed.
During the 7th day in ICU, the postoperative wounds were reviewed. Local evolution was favorable, fascial necrosis and pathological secretions were absent. Granulation tissue is present in the wound. These aspects led us to the decision to finally closing the wound (wound secondary sutures) and to suppress the remaining drain tube located into anterior mediastinum (Figure 3).
|
As the general health status did not improve, on the 19th day in ICU, it was decided to re-evaluate the case by performing a new CT scan of head, neck and chest. The CT scan revealed a fused retrosternal collection from the anterior cervical region having the following dimensions 11/45/73 mm (AP/LL/CC), thrombosis of the right internal jugular vein, bronchopneumonia (presence of condensation foci of all pulmonary lobes located subpleural) and also the presence of posterior basal atelectasis. Clinical laboratory test results showed a higher number of: white blood cells (leukocytosis), PCR and lactate dehydrogenase (WBC - 39.0 103/μl; PCR - 350 mg/L; LDH - 125 - 220). These clinical data led us to the therapeutic decision to review the postoperative right laterocervical wound and the anterior mediastinum is. A mixed team of maxillo-facial and thoracic surgeons was formed. During the surgical exploration of the anterior mediastinum, purulent collections were absent detected, only inflammatory infiltrate. The general health status of the patient worsened due to pulmonary hemorrhage complicated with bronchopneumonia.
On 25th day in ICU bronchial lavages were performed (for 5 days), with the removal of a blood clot formed in larger caliber bronchi (figure 4).
The general treatment of the patient was complex and was permanently adjusted to the curative needs and combining rheological treatment, Dopamine (positive ionotropic support), diuretics, hypotensive drugs, Clexane, hepatoprotective drugs, antibiotics, anti-inflammatory, analgesic, corticosteroid, vitamin therapy, fresh frozen plasma (FFP) and blood transfusion.
The initial antibiotic treatment during hospitalization was ceftriaxone and metronidazole. After the patient being transferred to the ICU, the initial antibiotic therapy was replaced using a combination of meropenem, vancomycin and metronidazole. At the ICU because of uncertain local evolution, various combinations of antibiotic treatment therapies were applied: meropenem, vancomycin and clindamycin; later clindamycin, gentamicin, imipenem and linezolid; and finally a combination of gentamicin, imipenem and linezolid. During the ICU hospitalization a nasogastric tube was placed for parenteral nutrition.
On 27th day in ICU, tracheostomy is performed due to need of prolonged mechanical ventilation and respiratory recovery.
On 34th day in ICU, general condition of the patient stabilized and slowly improved, confirmed by the improvement of the clinical laboratory test results. Final CT scan of the head, neck and thoracic cavity excluded any other pathologies. (Figure 5). Extubation and tracheostomy tube was removed, and the patien had adequate spontaneous breathing.
Figure 10: Final Ct scan showing healing of the head, neck and mediastinum, with no further pathologies.
Point 2: In line 279-280, the presence of Streptococcus anginosus and Granulicatella adiacens is mentioned. It would be helpful to provide information on whether any specific treatments were applied to target these two pathogens. Elaborating on the treatment approach for these specific pathogens would enhance the understanding of the therapeutic strategy employed.
Response 2: The bacteriological examination showed the presence of Streptococcus anginosus and Granulicatella adiacens. The presence of Granulicatella adiacens to the cervical level can explain the rapid evolution of the case from submandibular abscess to necrotizing fasciitis within a few hours of admission. Granulicatella adiacens has been described by others studies as being involved in endocarditis infections, breast or peritoneal implants.
Streptococcus anginosus colonizes the mucous membranes of the oral cavity and in some cases blood stream infections. Particularity of this case is that despite of the combined wide spectrum intravenous antibiotic treatment administered from the day of the admission (ceftriaxone and metronidazole) propagation and dissemination of the septic process was extremely fast. In approximately 48 hours it diffused from the left submandibular space towards the contralateral side, infratemporal fossa and the mediastinum. At this moment we did not dispose of the result of the bacteriological exam. Theoretically Streptococcus anginosus is susceptible ceftriaxone and metronidazole combination. Granulicatella adiacens is only partly susceptible to this antibiotic combination, and the synergic activity of these two aggressive strains of bacteria might have contributed to the fullminant evolution of the septic process. Dental infections are known to be polymicrobial, and there were no bacteriological exam for anaerob strains made, so this aspect of the investigations had limitations.
In cases of cervico-facial infections of the deep regions, the CT scan is the only reliable paraclinical exam which can show the presence of pus collections, necrosis (gases) in theese areas. Clinical signs are always blurred by the presence of the soft cervico-facial tissues, in particular in case of obese children having “short neck” and abundant cellulo adippose soft tissue.
For cervicofacial necrotizing fasciitis, the treatment of choice is surgical and it is a major emergency because it is a rare, fatal, rapidly progressive soft tissue infection. After extensive surgery, antimicrobial treatment was changed to meropenem, vancomycin and metronidazole. This combination should have covered all the bacterial strains involved in the infection. It would have been optimal to have targeted antibiotic therapy from the beginning, but to obtain bacteriological exam results is time consuming, and in these cases empiric antibiotic therapy is initiated, which is later adjusted to the bacteriological exam. In this case it consisted of antibiotic combinations such as clindamycin, gentamicin, imipenem and linezolid; and finally a combination of gentamicin, imipenem and linezolid
Point 3: The authors should expand the conclusions section to provide a clearer understanding of the lessons learned from this case. It would be valuable to highlight any efficient methods or interventions utilized during the treatment of this case that could be applicable to future cases. This expansion would provide practical insights for clinicians and researchers in similar scenarios.
Response 3: Oral cavity infections can have a fulminant evolution, having a rather serious prognosis as life-threatening development. In case of children most common odontogenic cause is dental decay complicated with pulp gangrene and periapical abcess. Rapid development, particularly in case of the infection to the deep perioral spaces such as the submandibular, sublingual, parapharyngeal, pterigomaxillar spaces is usually possible from the posterior permanent teeth, most frequently from the first and second molars, which erupt at the age of 6 yo., respectively at the age of 12 yo. In case of children, young molars have long roots, inserted deep in the jaws, and large root canals, with wide apical foramen, permit rapid development of the deep periapical abcess. Usually dental treatment of these molars is laborious, due to the poor collaboration with children. Silent oligosimptomatic dental pulp necrosis and gangrene under large dental fillings is also common in case of children due to large dental pulp chamber in case of recently erupted molars, which lay in the proximity of the dental filling material. Necrosis and gangrene can occure also due to cytotoxic properties of the dental filling materials. Without dental x ray, and thorough dental examination there can no correct dental emergency treatment be implemented. In most cases a dental trepanation at the visit to the dental doctor can prevent the later complications. Antibiotic treatment without proper dental treatment contributes to the selection of multi resistant anaerobic strains.
Invovement of the deep perioral spaces such as the submandibular space causes severe trismus, which obstructs any other dental treatment for the period of its persistance. Endooral examination is also limited due to intense trismus, poor collaboration with children also makes the treatment more difficult.
In order to avoid difficult intubation with potential risks, submandibular abcess incision with Hilton tehnique is made in sedation, which permits only limited surgical access and exploration time, and can lead to insufficient drainage of the affected space, and limited treatment possibilities. Bacteriological exam result is time consuming and in case of rapid, unfavorable, extensive infections, with no tendencies toward abcess collection, and limmitation to a single anatomic space, wide spectrum intravenous asssociated empiric antibiotic treatment must be initiated.
Usually children due to lack of understanding, cannot describe exactly the subjective simptomatology. Deep cervico-facial infection remain clinically masked for longer periods of time, in particular in case of children, where anatomy of the head and neck and specific landmarks are different than in adults. Obesity and abundent cellulo-adippose soft tissues blurr the clinical image even more. CT scan of the head, neck and even the torax in case of severe infections is the most reliable imagistic procedure that can indicate the emergency surgical treatment. Surgical treatment has to be extensive and control Ct scan should be made in order to confirm the exploration and the drainage of the involved anatomic spaces.
Until the result of the microbiologic exam empirically used combination of wide spectrum antibiotic therapy, and later targeted antibiotic therapy should be implemented.
Leaving the surgical wound open for longer periods of time, oxigenation of the involved deep anatomic regions, can assure effective treatment against anaerobic strains. Repeted necrectomy, consisting of the excision of fascias can lead to lesion of the great vessels of the head and neck, leading to uncontrollable bleeding.
Periodic reevaluation of the local and regional evolution, brain edema, brain abcess should be excluded on repeted Ct scans.
Long intubation period can cause ventillation assisted bronchopneumonia, aspiration pneumonia, and negative pressure edema and hemorrhage, specifically in case of young adults.
Ketoacidosis and septic shok, contribute to pressure ulcers in case of prolonged bedrest
Multidisciplinary treatment has to be implemented, regarding the immediate and the aftertreatment.
Point 4: I recommend that the authors thoroughly review the captions for all figures in the manuscript. I noticed instances of incomplete content in some captions, such as Figure 8. Furthermore, there was an error in figure naming, as mentioned in line 229, where it should be Figure 6 instead of Figure 1. Reviewing and revising the figure captions will ensure accurate and comprehensive descriptions of the figures, improving the overall clarity of the manuscript.
Response 4: Changes have been made in this regard, on the rewieved uploaded version

Reviewer 2 Report
Several questions:
The authors should explain in more detail how the mediastinal drainage was performed. As they write it, it is spectacular, but they do not detail the surgical procedure performed.
How was the drainage performed with sclerosis of the common carotid artery and its internal and external branches, internal jugular vein, right next to the right brachiocephalic artery? Were all these vessels ligated? What impact did this procedure have on the patient?
Line 114, 145, 151...: keep the same verb tense. Do not alternate present with past.
Pressure ulcers are potentially preventable by measures such as postural changes. How do the authors explain the occurrence of multiple severe ulcers in this patient?
The wording of the clinical case is too long; it should be synthesized and shortened, giving more internal cohesion to the text.
The authors should adapt the verb tense of the clinical case to the simple past. They should correct the numerous errors throughout the manuscript in this regard. We recommend a reading and grammatical correction by a native speaker.
Author Response
Point 1: The authors should explain in more detail how the mediastinal drainage was performed. As they write it, it is spectacular, but they do not detail the surgical procedure performed.
Response 1: Drainage of the anterior superior mediastinum was performed, entering from the neck incision, dissecting bilaterally along the anterior border of the SCM muscle, incising the superficial cervical fascia, and both layers of the deep cervical fascia above the jugular notch. Keeping contact with the internal surface of the manubrium sterni, blunt dissection was made, approximately 12 cm long, caudally. Necrotic secretions were evacuated, and drainage was achieved with silicone tube
Point 2: How was the drainage performed with sclerosis of the common carotid artery and its internal and external branches, internal jugular vein, right next to the right brachiocephalic artery? Were all these vessels ligated? What impact did this procedure have on the patient?
Response 2: There has been an auto correction error, we corrected it. it is about skeletonizations of the great vessels of the neck.
with skeletonization of the common carotid artery and its internal and external branches, internal jugular vein, right next to the right brachiocephalic artery.
Point 3: Line 114, 145, 151...: keep the same verb tense. Do not alternate present with past.
Response 3: verb tenses have been corrected, and we will ask for major revision of language from a native speaker.
Point 4: Pressure ulcers are potentially preventable by measures such as postural changes. How do the authors explain the occurrence of multiple severe ulcers in this patient?
Response 4:
Although specific mobilization was made in every 4-5 hours, and anti bedsore mattress was used, due to the peripherial vascular damage, aggravated by metabolic ketoacidosis and septic shok, pressure ulcers have evolved.
Point 5: The wording of the clinical case is too long; it should be synthesized and shortened, giving more internal cohesion to the text.
Response 5: A 12-year-old boy from urban area presented to Dental Emergency Room from Emergency County Hospital Târgu Mures, complaining of discomfort and pain located at tooth 3.7. The initial diagnostic was congestive pericoronitis at tooth 3.7. Painkillers, anti-inflammatory drugs were prescribed and local irrigations were performed using antiseptic solution under the mucosal cap covering the tooth 3.7. After two days he presented again to the Dental Emergency Room, complaining of pain in the left side of the mandible, pain that could not be accurately located and this time was diagnosed with suppurative pericoronitis at tooth 3.7. Local irrigation with chlorhexidine gluconate 0.2% was performed and peroral antibiotic treatment was prescribed (Amoxicillin tablets). After 12 hours there was a worsening of the general state of health of the patient, increased pain of the left side of the face and swelling of the left submandibular regoin. The patient was admitted in emergency to the Oral and Maxillofacial surgery department with the diagnostic of left submandibular abscess. Extraoral examination revealed facial asymmetry due to the swelling of the left submandibular and submental region and the skin over the swelling showed a local hyperemia. On palpation, the swelling was painful, firm in consistency, and without fluctuation. Thorough examination of the oral cavity was difficult to perform at that time, due to the presence of the trismus, but there was significant edema on the anterior and left side of the floor of the mouth present . A general physical examination was performed. Class 1 obesity, oscillating blood sugar levels and increased blood pressure was noticed, but without any other previously diagnosed general illness. The emergency treatment performed in local anesthesia consisted in incision, evacuation and drainage of the left submandibular abcess. Dissection of the left submandibular space was prformed using Hilton technique. Drainage tubes were positioned and sutured in the affected anatomical space. Despite the emergency surgical treatment and the assotiated intravenous antibiotic medication treatment initiated (Ceftriaxone and Metronidazole), the evolution of the patient status was unfavorable. 24 hours after the first surgical intervention left submandibular edema expansion towards the left submental area, towards the midline and infiltration of the contralateral submandibular and submental spaces, swallowing difficulties and worsening of the patient's general condition suggested a second surgical exploration of these spaces. Right submandibular and submental spaces were drained and bacteriological examination was requested. During the second intervention, the presence of gas in the right submandibular region is noticed with the presence of necrotic and fetid odor secretion. The patient was transferred to the ICU (Intensive Care Unit). After two hours from the transfer the general health status of the patient worsened. The patient showed signs of psychomotor agitation and became uncooperative and was sedated and intubated. Postoperative head, neck and thorax CT scan was performed in order to evaluate the extension of the lesions. The contrast-enhanced CT scan of head, neck and thorax revealed multiple mixed fluid-air densities in the sublingual, submandibular, submental, parapharyngeal, retropharyngeal, peripharyngeal spaces and having extension to the mediastinal level, on the upper and middle floors of the anterior and posterior compartment; also revealed edematous infiltration of peri-, submandibular and laterocervical subcutaneous soft tissues bilateral (Figure 1)
In the same day surgical reintervention was performed by joining and widening the bilateral subangulomandibular incisions, the submandibular, sublingual, the base of the tongue and bilateral infratemporal fossae were opened, explored by means of blunt dissection. Necrotic secretions and fetid-smelling gases were eliminated. A right curved horizontal laterocervical incision was performed to open the vascular space of the common carotid artery and its branches, the internal jugular vein, the paratracheal space (anterior, posterior and right lateral), reaching also to the right prevertebral space. Drainage of the anterior superior mediastinum was performed, entering from the neck incision, dissecting bilaterally along the anterior border of the SCM muscle, incising the superficial cervical fascia, and both layers of the deep cervical fascia above the jugular notch. Keeping contact with the internal surface of the manubrium sterni, blunt dissection was made, approximately 12 cm long, caudally. Necrotic secretions were evacuated, and drainage was achieved with silicone tube. All the anatomic spaces described previously were meshed. After the surgery, the treatment continued in the children's intensive care unit, the patient remained sedated, intubated and mechanically ventilated. The wound dressings were changed 3 time/day. During daily wound care surgical debridement, necrotic fascia mainly around the great vessels of the neck were excised(with skeletonization of the common carotid artery and its internal and external branches, internal jugular vein, right next to the right brachiocephalic artery). Due to the complex local treatment, local evolution was slowly favorable.
Regarding the general status of the patient, during the first days spent in the ICU, the patient presented acid-base and hydroelectrolyte imbalances, having oscillating pH between 7.26 – 7.56 (n = 7.3 – 7.45). Bacteriological examination of secretions taken from intraoperative wounds revealed the presence of Streptococcus anginosus and Granulicatella adiacens.
48 hours after the third surgical intervention, control CT scan of head, neck and chest with contrast was made. It revealed a quantitative reduction of air densities in the laterocervical, submandibular, parapharyngeal and mediastinal spaces, but with the presence of parafluid collections in the cervical and mediastinal spaces having approximately the same aspect as in the previous CT scan; minimal bilateral postero-basal pleural fluid collection was observed.
Entering from the neck scar, blunt dissection was made from the right side of the neck in order to drain the prevertebral collection that descends into the posterior mediastinum and necrectomy of the necrotic tissues and fascia was performed.
On 4th day in ICU: signs of peripheral vascular damage - severe acrocyanosis and pressure ulcers in the calcaneus and legs regions were observed.
On the 5th day in ICU, during dressing of the wounds (hour 9:00 P.M.) a massive erosive hemorrhage from the right laterocervical postoperative wound was produced. The bleeding was stopped by applying direct pressure to the wound.
Because of this event a head, neck, chest CT angiography is performed and no source of active bleeding in the thoracic cavity was revealed.
During the 7th day in ICU, the postoperative wounds were reviewed. Local evolution was favorable, fascial necrosis and pathological secretions were absent. Granulation tissue is present in the wound. These aspects led us to the decision to finally closing the wound (wound secondary sutures) and to suppress the remaining drain tube located into anterior mediastinum (Figure 3).
As the general health status did not improve, on the 19th day in ICU, it was decided to re-evaluate the case by performing a new CT scan of head, neck and chest. The CT scan revealed a fused retrosternal collection from the anterior cervical region having the following dimensions 11/45/73 mm (AP/LL/CC), thrombosis of the right internal jugular vein, bronchopneumonia (presence of condensation foci of all pulmonary lobes located subpleural) and also the presence of posterior basal atelectasis. Clinical laboratory test results showed a higher number of: white blood cells (leukocytosis), PCR and lactate dehydrogenase (WBC - 39.0 103/μl; PCR - 350 mg/L; LDH - 125 - 220). These clinical data led us to the therapeutic decision to review the postoperative right laterocervical wound and the anterior mediastinum is. A mixed team of maxillo-facial and thoracic surgeons was formed. During the surgical exploration of the anterior mediastinum, purulent collections were absent detected, only inflammatory infiltrate. The general health status of the patient worsened due to pulmonary hemorrhage complicated with bronchopneumonia.
On 25th day in ICU bronchial lavages were performed (for 5 days), with the removal of a blood clot formed in larger caliber bronchi (figure 4).
The general treatment of the patient was complex and was permanently adjusted to the curative needs and combining rheological treatment, Dopamine (positive ionotropic support), diuretics, hypotensive drugs, Clexane, hepatoprotective drugs, antibiotics, anti-inflammatory, analgesic, corticosteroid, vitamin therapy, fresh frozen plasma (FFP) and blood transfusion.
The initial antibiotic treatment during hospitalization was ceftriaxone and metronidazole. After the patient being transferred to the ICU, the initial antibiotic therapy was replaced using a combination of meropenem, vancomycin and metronidazole. At the ICU because of uncertain local evolution, various antibiotic treatment therapies were applied: meropenem, vancomycin and clindamycin; clindamycin, gentamicin, imipenem and linezolid; gentamicin, imipenem and linezolid. During the ICU hospitalization a nasogastric tube was placed for parenteral nutrition.
On 27th day in ICU, tracheostomy is performed due to need of prolonged mechanical ventilation and respiratory recovery.
On 34th day in ICU, evolution of the case is favorable. The general state of health has improved, also improvement of the clinical laboratory test results were observed. Due to theese improvements, extubation and tracheostomy tube removal it is decided, having adequate spontaneous breathing.
On 39th day, he is transferred to the OMF Surgery Department for further specialized treatment. Due to prolonged immobilization, the patient had also pressure ulcers (occipital, legs and calcaneus) having signs of healing as the presence of granulation tissue and scarring (Figure 5 and 6).
A Pediatric Neuropsychiatry consult was also performed after which the diagnosis of acute flaccid paralysis and anxiety disorder was established. When the general state of health was stable, an OPG was performed and on the tooth 3.6 a deep cavity, damaging the pulp chamber was observed and a periapical reaction was evident at the distal root (Figure 7-a). Intraorally, we observed that 3.7 have the persistence of a mucous cap at the coronary level - distal part (Figure 7-b). It was decided to extract tooth 3.6 and curettage the periapical lesion (Figure 7-c).
On 48th day, the patient was discharged having following recommendations: kinetic therapy, specific treatment according to recommendations of Pediatric Neuropsychiatry, dressing of all pressure ulcers, post-tracheostomy wound and periodic check-ups. After 2 weeks of discharge from the hospital spontaneous closure of the tracheostomy wound and postoperative scars on the neck can be observed (Figure 8).
Reviewer 3 Report
The article “Pediatric Cervicofacial Necrotizing Fasciitis - a challenge for the medical team” is a case report very interesting and I have some comments to make to improve the article.
- It would be more appropriate to write “Operculum” than mucous cap
- Did the patient have any systemic disease? Do you have a previous x-ray or image?
- What was the antiseptic solution you used (line 73)?
- It would be more appropriate to write “Operculum” than mucous cap
- Did the patient have any systemic disease? Do you have a previous x-ray or image?
- What was the antiseptic solution you used (line 73)?
- In the first days, did you take a bacteriological sample from the area around the operculum?
- What was the first antibiotic you prescribed? Line 77, and Later in Line 198 you write "initial antibiotic treatment was ceftriaxone and metronidazole"
- Line 199-202. It is a bit confusing regarding the sequence of antibiotics administered
- On which day was the patient's general condition stable?
- Was the periapical lesion of 3.7 analyzed histopathologically?
- Was an operculectomy performed?
- Of the cases described in the literature of Necrotizing fasciitis, which is the most frequent odontogenic cause? Possible causes should be mentioned in the discussion
- What antibiotic did you propose, or what is it that you propose for the reader in a preventive way in the presence of Streptococcus anginosus and Granulicatella adiacens?
Thank you
Author Response
Response to Reviewer 1 Comments
Point 1 - It would be more appropriate to write “Operculum” than mucous cap
Response 1: Changes have been made in the manuscript, mucosus cap has been changed to operculum
Point 2 - What was the antiseptic solution you used (line 73)?
Response 2: chlorhexidine gluconate 0.2%
Point 3: - Did the patient have any systemic disease? Do you have a previous x-ray or image?
Response 3: A general physical examination was performed. Class 1 obesity, oscillating blood sugar levels and increased blood pressure were noticed, but without any other previously diagnosed general illness. We had no previous dental Xrays in the moment of admission. Ct scan of the head, neck and thorax was made, in order to evaluate de progression of the fasciitis
Point 4: - In the first days, did you take a bacteriological sample from the area around the operculum?
Response 4: No bacteriological sample was taken from the area around the operculum due to intense trismus which permitted no access to the oral cavity. Bacteriologicla sample was taken from the submandibular secretions after the incision has been preformed.
Point 5: - What was the first antibiotic you prescribed? Line 77, and Later in Line 198 you write "initial antibiotic treatment was ceftriaxone and metronidazole"
Response 5: The first antibiotic prescribed was Amoxicillin tablets by the dentist. After the admission to the hospital, combined iv antibiotic tratment was administered, ceftriaxone and metronidazole
Point 6:- Line 199-202. It is a bit confusing regarding the sequence of antibiotics administered
Response 6: . At the ICU because of uncertain local evolution, various combinations of antibiotic treatment therapies were applied: meropenem, vancomycin and clindamycin; later clindamycin, gentamicin, imipenem and linezolid; and finally a combination of gentamicin, imipenem and linezolid.
Point 7: - On which day was the patient's general condition stable?
Response 7: On 34th day in ICU, general condition of the patient stabilized and slowly improved, confirmed by the improvement of the clinical laboratory test results. Extubation and tracheostomy tube was removed, and the patien had adequate spontaneous breathing.
Point 8: - Was the periapical lesion of 3.7 analyzed histopathologically?
Response 8: No histopathologic exam was made from the 3.7 periapical region
Point 9 : - Was an operculectomy performed?
Response 9: Intraorally, we observed that the operculum of tooth 3.7 was partly present on the distal occlusal surface. It was decided to extract tooth 3.6 and curettage the periapical lesion
Point 10:- Of the cases described in the literature of Necrotizing fasciitis, which is the most frequent odontogenic cause? Possible causes should be mentioned in the discussion
Response 9: Necotizing fasciitis is rare in the head and neck region, but it is potetntially life threatening, most frequently caused by the second and third lower molars in case of adults [33], it can appear in case of children, caused by the periapical pathology of the first molar.
The clinical examination is difficult to perform in case of a child, due to tension that can be generated by the small patient's meeting with the doctor. The anamnesis is most often inconclusive, because the child has difficulty expressing the painful sensations and cannot accurately specify their location. This makes it difficult for the doctor to establish the correct diagnosis. In group age between 6-12 years, takes place dental eruption of permanent molars. Development of dental caries during this period and their rapid progression to the pulp chamber and which can be complicated with pulp gangrene catches the tooth having incomplete root formation and root development is stopped. Consequently, the root canal is large, with thin and fragile walls, with open apex [[i]]. And this is a quickly exit gate for bacteria to the surrounding spaces. Eruption of the molars can also be associated with another type of pathology, namely pericoronitis (inflammation of the mucous membrane above the molar). The two associated pathologies can lead to multiplication of bacteria, exacerbation of pain and implicitly a much faster diffusion to neighboring anatomical spaces (submandibular, sublingual, submental, prevertebral, parapharyngeal, and retropharyngeal spaces).
In our case, we consider the pericoronitis of the lower left second molar as the starting point of the infection. The accumulation of bacteria under the mucous cap could go unnoticed due to the vague symptomatology at the time of presentation to the dentist.
Complicated caries and dental gangrene are the causes of two-thirds of cases, followed by pericoronitis and periodontal disease. Tonsil infections, salivary gland infections, otogenic and dermatological infections are other causes.
Point 11: What antibiotic did you propose, or what is it that you propose for the reader in a preventive way in the presence of Streptococcus anginosus and Granulicatella adiacens?
Response 10: - Necrotizing fasciitis as a complication of odontogenic infection is polimcrobial and a non specific infection, Usually, lacking the bacteriological exam result, in emergency conditions, it is recommended to use wide spectrum antibiotics, sometimes combinations of theese to cover the anaerob microbes also.
Round 2
Reviewer 1 Report
The revised manuscript has addressed all my concerns.
Minor editing of English language required
Author Response
Point 1. Minor editing of English language required
Response 1: Thank You for Your help. The revised version of the manuscript is attached

Reviewer 2 Report
The authors have answered most of the questions raised.
However, two additional issues need to be addressed before publication. A review of the English language by a native speaker. The wording of the clinical case is still too long; it should be synthesized and shortened, giving more internal cohesion to the text.
The authors have answered most of the questions raised.
However, two additional issues need to be addressed before publication. A review of the English language by a native speaker. The wording of the clinical case is still too long; it should be synthesized and shortened, giving more internal cohesion to the text.
Author Response
Point 1: However, two additional issues need to be addressed before publication. A review of the English language by a native speaker. The wording of the clinical case is still too long; it should be synthesized and shortened, giving more internal cohesion to the text.
Response 1: Review of the English language of the paper has been made. We tried to shorten the wording of the clinical case as much as possible. We think that the detailed description of this case is essential for the better understanding of the rapid evolution and complexity of this case.
